# Herd Health Troubles Potentially Related to Aluminium Grass Silage Content in Dairy Cows

**DOI:** 10.3390/vetsci10020149

**Published:** 2023-02-12

**Authors:** Justine Eppe, Salem Djebala, Frédéric Rollin, Hugues Guyot

**Affiliations:** 1Clinical Department of Production Animals, Fundamental and Applied Research for Animals & Health Research Unit (FARAH), Faculty of Veterinary Medicine, University of Liège, Quartier Vallée 2, Avenue de Cureghem 7A-7D, 4000 Liège, Belgium; 2Murphy and Leslie Veterinary Center (Private Practice), Muckerstaff Granard, N39AN52 Co Longford, Ireland

**Keywords:** poisoning, cattle, metabolism, nutrition, roughages

## Abstract

**Simple Summary:**

Aluminium intoxication is poorly documented in ruminants and its symptomatology, compatible with grass tetany caused by hypomagnesaemia, is the most documented clinical manifestation of acute intoxication. However, there is no documented evidence of chronic aluminium intoxication in cattle. In a 50 dairy cow Belgian herd, excessive uterine bleeding at calving and decreased milk production were reported. The results of various analyses on 10 sick cows were compared with 10 healthy cows from another herd. The investigations of sick animals showed anaemia, marginal hypozincaemia, subclinical ketosis, hypomagnesaemia and a high aluminium/creatinine ratio (urinary excretion). The mixed ration contained a high level of aluminium. Based on the results and suspicion of chronic aluminium poisoning, it was advised to measure the soil pH, add salts to the ration to chelate the aluminium and support the cows with mineral supplements and propylene glycol. A visit was carried out 2 years later. The situation had improved, but all of the cows examined had subclinical ketosis. The grass silage had high aluminium and butyric acid concentrations. Aluminium could be incriminated in different stages, but it was probably not the only culprit. Chronic poisoning with metals and pollutants should be the focus of veterinary attention in the coming years.

**Abstract:**

In ruminants, the main documented clinical manifestation of aluminium (Al) intoxication is similar to grass tetany. In a 50 dairy cow Belgian herd, the farmer reported excessive uterine bleeding at calving and decreased milk production. Dairy cows received a mixed ration (MR) with high Al concentration (453 ppm/kg of dry matter (DM)). Various analyses were sampled from 10 sick cows and compared with 10 healthy cows (from another herd). Sick cows presented anaemia and marginal hypozincaemia and 6/10 showed subclinical ketosis. Their urine analysis revealed hypomagnesaemia and a high Al/creatinine ratio. It was advised to determine soil pH, add salts to the ration to chelate the Al and support cows with mineral supplements and propylene glycol. A visit was carried out 2 years later and highlighted an improvement in the situation, but all examined animals presented subclinical ketosis. Grass silage Al content remained high (700 ppm/kg DM), as did butyric acid concentration (11.22 g/kg DM). Al could be incriminated at different stages: micronutrient deficiencies, anaemia and negative energy balance. However, Al was probably not the only culprit. This case report is a concern for future years in these areas due to droughts, scarcity of forage and an increase in contaminated soil ingestion.

## 1. Introduction

Aluminium (Al) is the third most common element on Earth. Its bioavailability increases in acid soils (pH < 5.5) [1]. Once Al is ingested, it disrupts the absorption of various elements such as magnesium (Mg), phosphorus (P), calcium (Ca), zinc (Zn) and iron (Fe) [2,3,4]. Al is also able to stabilize Fe^2+^, preventing its oxidation into Fe^3+^ [5].

There is abundant literature on Al intoxication in humans, through food and drinking water but also via the administration of drugs or vaccines containing Al-based adjuvants [6,7,8]. The potential risks associated with exposure to Al depend on its form, duration, frequency, dose and administration route [8]. The pathogenesis of Al is explained by its very good absorption in all organs, such as the brain, liver, lungs, bones and muscles [5]. This leads to a wide range of symptomatology, such as changes in the evolution of secondary hyperparathyroidism, leading to Al-induced osteomalacia, alterations in Fe metabolism [6] with microcytic anaemia, a sclerosing effect on nerve cells, diseases such as Alzheimer’s or Parkinson’s [9] and numerous other clinical manifestations. Because of its various indirect effects, the diagnosis of Al intoxication is performed by exclusion, and has no pathognomonic clinical picture.

In mice, long-term exposure to Al mimics liver senescence, ultimately causing liver damage [9]. One study has shown fenugreek to be effective as a liver protector during prolonged exposure to Al, suggesting its use in farm animals [10], although no such effects have been documented in cattle to date. 

Al intoxication in ruminants is not well documented. There are different sources of Al for ruminants [2,11], such as ingestion of soil-contaminated roughages, ingestion of forages containing Al accumulator plants (for example, *Camellia sinensis*, *Hydrangea macrophylla* and *Fagopyron esculentum*), contaminated water [12,13,14] and Al-based additives. Ruminants could also be contaminated in pasture, which is the more plausible as ruminants are known to ingest soil. This ingestion can vary up to 18% of the dry matter (DM) ingested when grass is scarce (drought, early spring, late summer) but decreases to 2% of the DM ingested when grass is at a good height [15,16]. The most documented clinical symptom of Al intoxication is grass tetany, caused by hypomagnesaemia [3,11]. Nowadays, no other Al clinical signs have been reported in cattle. These intoxications are acute, following the ingestion of Al doses higher than 1000 ppm [2,17]. Nevertheless, it is not known whether the ingestion of lower doses (range 300 to 1000 ppm) can provide a negative chronic influence on cattle [17]. This case report is displayed in this context, aiming to warn about the potential effects of a chronic Al ingestion in dairy cows.

## 2. Case History

On a farm located in Trooz, cradle of the textile, agro-alimentary and metallurgy industries in the province of Liege (Belgium; Figure 1) [18], with 50 milking dairy cows, a farmer reported uterine bleeding at calving and suboptimal performances (from 8000 L to 7000 L of milk/cow/year, according to milk recordings). The prevalence of these troubles was estimated to be around 20% of the milking cows. 

## 3. Methodology

Examinations were performed on 10 randomly selected sick dairy cows (sick pool). A blood sample was taken from each of these cows at the level of the coccygeal vein (vacutainer©; 18G needle, BD Benelux, Erembodegem, Belgium). The estimated fibrinogen concentration was calculated as the difference between protein concentrations in the serum and plasma measured by optical refractometry (Refractometer, Euromex^®^, Arnhem, The Netherlands). Haematocrit and haemoglobin were measured using a portable analyser (Mission-Hb haemoglobinometer^®^, Acon, San Diego, CA, USA). Ketosis was assessed via the determination of blood beta-hydroxybutyrate (BHB; Freestyle Precision ß Ketonen Teststrips, Abbot, Chicago, IL, USA). Serum zinc (colorimetric test, Iodolab, Grézieu-la-Varenne, France) and urinary magnesium (Merkognost^®^, Merck kGaA, Darmstadt, Germany) were assayed. In view of the geographical context and the clinical picture, urinary creatinine (Jaffe method on Cobas 8000, Department of Clinical Chemistry, University Hospital of Liège) and Al (inductively coupled plasma-mass-spectrometry, ICP-MS, Department of Clinical Chemistry, University Hospital of Liège) were added to the analyses. To minimize the effects of dehydration or any other physiological change, urinary Al was expressed as a ratio with creatinuria. Results of these investigations were compared to a population of 10 healthy cows from another dairy farm, unexposed to Al (ULiege pool). The semi-complete mixed ration (MR) was evaluated using the Larelev 2019 software to calculate the UFL (Unité Fourrage Lait) and PDI (Protéines Digestibes Intestinales) values of the ration. A careful analysis of the distributed, ingested, digested and metabolised ration was also performed. A mineral analysis of the ration and a content analysis of the grass silage were carried out (inductively coupled plasma–optical emission spectrometry, ICP-OES, Iodolab, Grézieu-la-Varenne, France). Rumen fill (RF; scale 1–5) [19,20], undigested fraction (UFS; scale 1–5), body condition score (BCS; scale 1–5) and faecal consistency (FCS; scale 1–5) scores were evaluated. To compare the two pools (“Sick” and “ULiege”), a *t*-test was performed using R software (version 4.2.1). Statistics were only carried out on parameters for which an individual value per animal was available. No statistics were carried out on the Al/Creat ratio and on blood Zn and Al because these were pooled samples, nor on the Al, Fe, Zn or Mg concentrations in the basic ration.

## 4. Results of the Investigations

The 10 randomly selected cows in the “Sick pool” had a general examination within the norms overall, with the exception of their pale mucous membranes. They showed a weak body condition [21] (Figure 2) but no inflammation (fibrinogen < 6 g/L) or hypoproteinaemia (range 57–81 g/L) [22]. However, they presented anaemia (haematocrit, Hct, 23 ± 3% (mean ± SD), threshold 22–33%), low haemoglobin (Hb, 7.9 ± 1.2 g/dL, norm 8.5–12.2 g/dL) and marginal Zn deficiency (pool value 12 µmol/L; norm 14–21 µmol/L; Iodolab, Grézieu-la-Varenne, France), and four out of ten cows showed subclinical ketosis (BHB > 1.2 mmol/L; norm < 1.2 mmol/L) [23]. Urine analysis revealed hypomagnesaemia (50 ± 29 mg/dL; norm 120–250 mg/dL) [17] and high Al/creatinine levels (Al/Creat = 75) in the sick pool compared to the ULiege pool (Al/Creat = 18) (Table 1). 

The semi-complete mixed ration (MR) was calculated for a production of 26 L of milk per day (basic ration). Dairy cows were fed first-cut grass silage (38 kg fresh/cow) and an energy corrector containing wheat, maize, barley (4 kg/cow) and over-pressed beet pulp (17 kg fresh/cow). In addition to the MR, a concentrate (18% crude protein) containing alfalfa was added for dairy cows (3–4 kg/cow) and for dry cows (0.5 kg/cow) in an automatic distributor. A balanced complement of production was distributed in the milking parlour to the dairy cows at a rate of 0.5–0.6 kg per cow depending on the milk production. The MR had the following characteristics in terms of energy and protein: 0.87 UFL/kg DM; 85 g PDI/kg DM; 97 g PDI/UFL. RF, UFS and FCS scores were within the norm (mean RF = 3; mean UFS = 2; mean FCS = 3). The only abnormal parameter observed was the BCS of the cows (Figure 2; 6/10 below the expected values). The farmer produced his own grass silage from his meadows. A macroscopic examination of the silage and meadows did not reveal any Al accumulator plants (for example, *Camellia sinensis*, *Hydrangea macrophylla* and *Fagopyron esculentum*) [12,13,14]. The grass silage was of average quality, with a measured DM content of 31% and 133 g/kg of ashes. The mineral analysis of the ration showed an unusual concentration of Al (453 ppm/kg DM; norm < 300 ppm), Fe to be in the high standards (492 ppm/kg DM; norm < 500 ppm) and normal Zn (92.6 ppm/kg DM; norm 50–100 ppm/kg DM) and Mg (0.23% of DM; norm 0.19–0.29% of DM) (Table 1) [17]. The ration of the ULiege pool was also analysed, and all of the above parameters (Al, Zn, Mg and Fe) were within the normal ranges.

Cows drank water from a well. Water was analysed and revealed no Al contamination (Al < 12 µg/L). 

Age and days in milk were therefore compared by means of this statistical test, as well as the blood determinations of serum total protein (STP), plasma total protein (PTP), fibrinogen (F), haematocrit (Hct), haemoglobin (Hb), BHB and the urine determination of Mg. Significant results (*p* < 0,05) were marked with a star in Table 1. Days in milk were significantly different (*p* < 0.05) between the sick pool and the ULiege pool, just as was the case with Hct, Hb, BHB and urinary Mg. The Al/creatinine ratio value of the sick pool is more than four times higher than that of the ULiege pool.

As there was a strong suspicion of Al intoxication, measures to limit this contamination were implemented. It was advised to determine the soil pH and to lime the grassland if the soil had an acidic pH. It was also advised to add salts to the ration, such as MgO, CaCO_3_ and NaHCO_3_, to induce precipitation of Al in the rumen, while being careful not to cause rumen alkalosis. To limit deficiencies, supporting cows with boluses enriched with minerals and trace elements (Mg, Zn and P) was advised. Moreover, administration of propylene glycol in the feedstuffs was also recommended to address ketosis. 

## 5. Follow-Up

A monitoring visit was carried out two years later. The farmer considered his situation to be improved, although he believed he still had suboptimal milk production (around 7000–7500 L of milk/cow/year, according to milk recordings). He was no longer complaining about bleeding during calving. The trace elements’ status was checked yearly, and animals with deficiencies were supplemented with boluses. The latest trace elements’ control (Zn, copper, iodine, selenium) and vitamin B12 has only showed a minor selenium (Se) deficiency (54 µg/L).

During this visit, cows (Table 2) revealed no abnormal values for haematocrit and plasmatic Zn but had low values of haemoglobin (6.3 ± 0.9 g/dL, norm 8.5–12.2 g/dL). Nevertheless, they had almost twice as many excretions of Al (Al/Creat) compared with the first visit. This was confirmed by the Al content of the grass silage that had still increased (700 ppm). The fodder analysis also revealed a high concentration of butyric acid (BA; 11.22 g/kg DM, threshold < 10 g/kg DM). Finally, all investigated animals presented subclinical ketosis (10/10, with blood BHB >1.2 mmol/L). The haemoglobin and BHB concentrations were significantly different (*p* < 0.05) between the sick and ULiege pool.

Equivalent recommendations to the first visit were made. It was additionally advised to analyse the Mg content of the urine of the cows after grazing, to monitor the deficiency. 

The farmer never followed advice directed towards checking the pH of the soil and adding salts to the ration. However, he had always monitored trace element balances to avoid deficiencies. The recommended trace element supplementation was sufficient to solve his problems. After this last visit, we were never called again for a problem at this farm.

## 6. Discussion

Al intoxication is widely documented in human medicine. It can lead to a very wide range of symptoms [5,8,25]. Poor documentation is available in ruminant species. Only clinical description such as grass tetany was reported [3,11]. Nevertheless, it is not known whether the long-term ingestion of lower doses than those documented in grass tetany can provide a chronic clinical picture in cattle. 

In this case report, no symptoms compatible with grass tetany were observed. However, Al could be involved at different levels, in particular with micronutrient deficiencies (including Mg), anaemia and negative energy balance. Al is known to interfere with the absorption of several elements (Mg, Zn and P). For example, P absorption is greatly affected as it bridges with Al to form the Al-P complex, which becomes unabsorbable and is eliminated by faeces [2]. The lack of P measurement could be one of this case report’s weaknesses. However, no haemoglobinuria that could be related to hypophosphataemia was found on this farm. A deficiency in other ions, due to the lack of intake in the MR, can be ruled out via micromineral analysis of the ration. In this case report, calculated MR seemed to cover all of the needs of dairy cows. The examination of the distributed, ingested and digested rations was not contributory.

Although the calculated MR appeared to be correct, a thorough examination of the metabolised ration was also carried out and revealed that 60% of tested cows had a BCS too low for their stage of lactation (Figure 2). This finding cannot be explained by the MR which seemed to be balanced. This clinical case involves two hypotheses, namely, whether the level of butyric acid or Al in grass silage could decrease DM consumption or increase requirements for cows in the case of chronic Al poisoning.

Anaemia may be the consequence of coagulation troubles due to hypozincaemia and hypomagnesaemia [26,27,28], themselves being a result of Al intoxication [2,3,4]. In our case, excessive bleeding at calving may be the consequence of the Mg deficiencies and marginal zincaemia. It is also known that Al can cause microcytic anaemia, either directly by acting on mechanisms of erythropoiesis, or through its effect of disrupting Fe absorption and its incorporation into red blood cells. This anaemia is similar to the one seen in Fe deficiency [5]. The reported anaemia was unfortunately not characterised because only haemoglobin and haematocrit were investigated using a portable device. The characterisation of this anaemia or the determination of serum Fe (low or normal serum Fe) could have confirmed the link between chronic Al intoxication and anaemia. 

There is no literature on chronic Al intoxication in ruminants. Liver damage, coming from an accelerated senescence, is described in mice chronically exposed to Al [9]. Furthermore, the liver appears to be an important storage organ for Al in steers [29]. The involvement of Al in the negative energy balance and subclinical ketosis in this case report is questionable. However, during the second visit, high BA (11.22 g/kg DM) levels were found in the silage, predisposing animals to type 3 ketosis. This finding was associated with 10/10 cows (advanced stage of lactation: 132 ± 85 days in milk) with a BHB > 1.2 mmol/L. The potential impact of Al in this mechanism is not known.

Although there is no standard for what is considered to be a normal urinary Al concentration in cattle, this case illustrates very different concentrations of urinary Al/Creat excretion in two different herds, living on two geographically different sites. Several studies have looked at the heavy metal content of cattle and dairy products. In these studies, serum levels of 189.4, 790, 277 and 1567 µg/L of Al are described [30,31], but nothing was specified about the health conditions of the enrolled animals. A serum concentration of 9–20 µg/L Al is taken as the standard in cattle [17]. In addition, serum Al does not reflect the amount of Al in an individual, because of its wide absorption by different tissues, except in the case of acute poisoning [7]. In our case of suspected chronic Al intoxication, it seems that urinary Al is a better indicator, because it reflects the diffusion of Al in the different organs of the body [32].

It made sense in this case to compare the sick population with a confirmed healthy population. Indeed, given the lack of literature on Al intoxication in ruminants, it is difficult to interpret values obtained without well-defined normal ranges. At the first visit, the age of the two populations was similar, but there was a significant difference in the days in milk. At the second visit, the two herds were comparable in age and days in milk. The values of days in milk at the first visit could explain the significant difference obtained for BHB while comparing the two herd visits.

In this case report, grass silage was clearly a suspect. A calculation of the Al concentration of the grass silage from the Al concentration of the MR (453 ppm/kg DM) and the proportion of grass silage in the MR revealed a grass silage concentration of nearly 725 ppm/kg DM, which corresponded to the concentration found in the second visit for grass silage (700 ppm/kg). This contamination is more likely due to soil contamination, given the Fe concentration (492 ppm/kg DM) and crude ashes (133 g/kg MS) in the grass silage, than to the presence of Al-accumulating plants [3,4,11]. This concentration could be decreased by liming the grassland if the soil had an acidic pH (pH < 5.5), because Al is absorbed by plants when the soil pH is acid [1,2]. The farmer never wanted to carry out this analysis, so we were unable to test this parameter.

The follow-up revealed an overall improvement in this farm. Nevertheless, the problem was not solved. On the contrary, the regular monitoring of the mineral concentrations of the cows has made it possible to adjust the supplements on a case-by-case basis. This management keeps the cows within the low average of the concentration standards for various minerals, which has allowed the suppression of several clinical symptoms. Finally, this seems to not be the only effect of Al, and a direct effect of Al on the liver cannot be ruled out; it might be the cause of the subclinical ketosis observed in this farm, since an effect on liver was previously reported in other studies and species [9,10].

There is a long history of industrialisation in the municipality of Trooz [18]. In the last decade, we have noticed various climate changes such as global warming [33] Climate change has direct effects on ruminants, such as heat stress, but also indirect effects through the quantity and quality of forage that may be available [34]. During the drought period, the concentration of Al in the soil associated with acidic soil can contribute to reduced plant growth [35], resulting in poorer forage production, which can lead farmers to cut the plant shorter and to risk contamination by soil. In addition to this, in July 2021, a flood inundated all of the valleys of the Ourthe and Vesdre rivers [36]. These floods could also be a source of contamination. All of these findings suggest that the environment in which ruminants live is changing, and that a greater prevalence of acute and chronic intoxication with different metals, hydrocarbons or factory wastes is to be expected. It is likely that cattle management will have to take the climate changes into account in the future.

## 7. Conclusions

Although many of the symptoms encountered in this case report could be attributed to Al, it may not be the only culprit, given the Fe and butyric acid contents of silage. This case raises the question of the involvement of chronic Al intoxication in animals. Further studies should be carried out on the potential effect of Al on other organs such as the liver in the long term. In future years, the consequences of climate change and pollution will have to be considered in the veterinary management of herds in our countries. 

## Figures and Tables

**Figure 1 vetsci-10-00149-f001:**
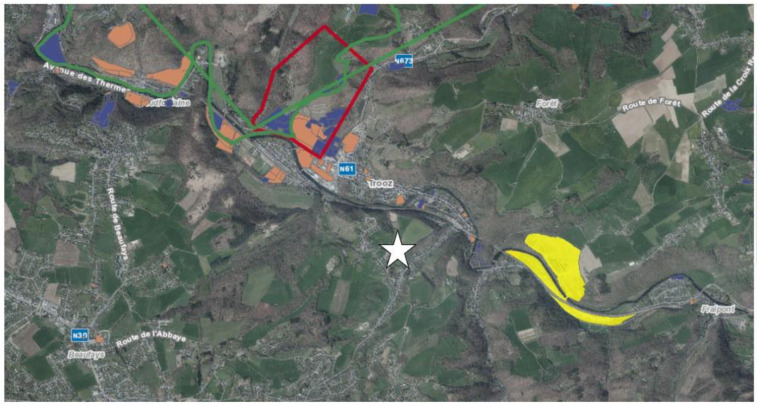
Geographic context of the farm (white star), according to the mining concessions, quarries and soil conditions (soil database: Géoportail de la Wallonie: https://geoportail.wallonie.be/home.html, Banque de Données de l’Etat des Sols—BDES; Service Public Wallonie—SPW, © Région wallonne, accessed on 7 November 2022). The green and red circled areas indicate coal or metal mining concessions, respectively. The yellow area indicates a siliceous rock quarry. The orange and blue zones are areas for which the pollution status is known. Orange areas require special treatment before building. The blue areas are provided as an indication of what was known about the industrialisation of this parcel. There are many coloured areas on the map because they correspond to areas of textile industrialisation.

**Figure 2 vetsci-10-00149-f002:**
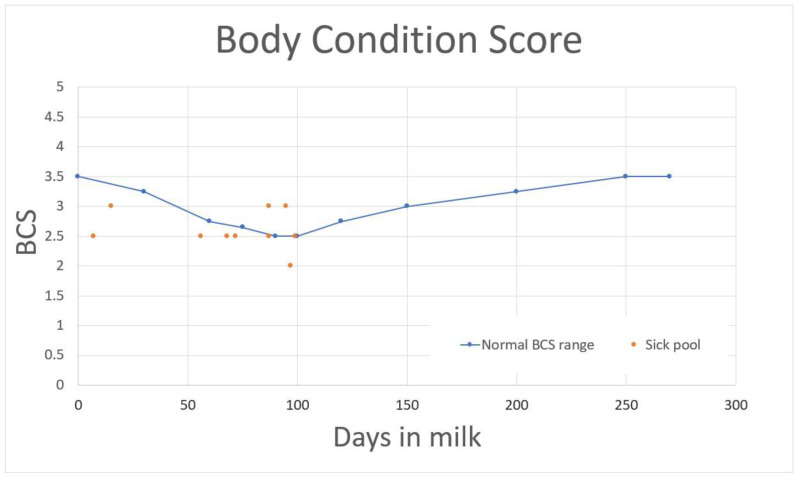
Body condition score (BCS) of the sick pool (red points) in comparison to the normal BCS range expected in relation to days in milk (blue line) [21].

**Table 1 vetsci-10-00149-t001:** Blood, urine and ration analysis (mean ± SD and pooled values) in the sick and ULiege (“healthy”) group of cows at the first visit.

Parameters (Unit)	Sick Pool	ULiege Pool	Threshold
GENERAL INFORMATION			
Age (years)	4 ± 1	3 ± 1.5	-
Days in milk	68 ± 33 *	192 ± 62	-
BLOOD			
STP (g/L)	72 ± 4	72 ± 9	57–81
PTP (g/L)	76 ± 3	75 ± 9	-
F (g/L)	4 ± 1	4 ± 1	<6
Hct (%)	**23 ± 3 ***	34 ± 3	23–33
Hb (g/dL)	**7.9 ± 1 ***	12.3 ± 1.1	8.5–12.2
Zn (µmol/L)	**12**	14	14–21
BHB (mmol/L)	**1.2 ± 0.5 ***	0.2 ± 0.1	<1.2
URINE			
Mg (mg/dL)	**50 ± 29 ***	130 ± 15	120–250
Al/Creat (-)	**75**	18	-
RATION WITH MINERAL SUPPLEMENT (/kg DM)			
Al (ppm)	**453**	105	<300
Fe (ppm)	492	382	<500
Mg (%)	0.23	0.27	0.19–0.29
Zn (ppm)	92.6	89	50–100

STP = serum total protein; PTP = plasmatic total protein; F = fibrinogen; Hct = haematocrit; Hb = haemoglobin (Mission-Hb haemoglobinometer^®^, Acon, San Diego, CA, USA); Zn = zinc; Al = aluminium; BHB = betahydroxybutyrate; Mg = magnesium; AI/Creat = Al to creatinine ratio; DM = dry matter; Fe = iron; in bold = non-standard results. Thresholds come from the laboratory and the literature [22,23,24]. * significant difference between sick and healthy herds assessed using *t*-test (*p* < 0.05).

**Table 2 vetsci-10-00149-t002:** Blood, urine and grass silage analysis (mean ± SD and pooled values) in the sick and ULiege (“healthy”) group of cows at the second visit.

Parameters (Unit)	Sick Pool	ULiege Pool	Threshold
GENERAL INFORMATION			
Age (years)	5 ± 2	6 ± 2	-
Days in milk	132 ± 85	180 ± 25	-
BLOOD			
STP (g/L)	72 ± 5	79 ± 5	57–81
PTP (g/L)	77 ± 5	82 ± 5	-
F (g/L)	5 ± 1	3 ± 2	<6
Hct (%)	28 ± 4	29 ± 2	23–33
Hb (g/dL)	**6.3 ± 0.9 ***	8.6 ± 0.9	8.5–12.2
Zn (µmol/L)	14.8	14	14–21
BHB (mmol/L)	**1.8 ± 0.7 ***	0.2 ± 0.1	<1.2
URINE			
Mg (mg/dL)	**83 ± 38 ***	130 ± 15	120–250
Al/Creat (-)	**145**	13	-
GRASS SILAGE (/kg DM)			
Al (ppm)	700	-	<300
BA (g/kg DM)	**11.22**	-	<10
Mg (%)	0.21	-	0.19–0.29

STP = serum total protein; PTP = plasmatic total protein; F = fibrinogen; Hct = haematocrit; Zn = zinc; Al = aluminium; BHB = betahydroxybutyrate; Mg = magnesium; AI/Creat = Al to creatinine ratio; Fe = iron; in bold = non-standard results. Thresholds come from the laboratory and the literature [22,23,24]. * significant difference between sick and healthy herds assessed using *t*-test (*p* < 0.05).

## Data Availability

All data relevant to the study are included in the article.

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
