# Peer review of "Herd Health Troubles Potentially Related to Aluminium Grass Silage Content in Dairy Cows"

_vetsci, 2023, doi:10.3390/vetsci10020149_

Round 1

Reviewer 1 Report

Interesting and well written paper.  

I could be acknowledged that the high silage  butyric acid may be from soil contamination.  The ash value of the silage would be helpful to determine this.  The Al may therefore be from the soil rather than the forage, so grass Al determination would also be of interest.  

The high butyric acid will also affect intake, which may be a contributory factor to the lowered milk yield, and also lowered mineral levels, leading to poor blood coagulation.

Minor correction: line 186, Vitamin B12 is technically not a trace element.

Author Response

REVIEWER 1

Comments and Suggestions for Authors

Interesting and well written paper.  

I could be acknowledged that the high silage  butyric acid may be from soil contamination.  The ash value of the silage would be helpful to determine this.  The Al may therefore be from the soil rather than the forage, so grass Al determination would also be of interest.  

The high butyric acid will also affect intake, which may be a contributory factor to the lowered milk yield, and also lowered mineral levels, leading to poor blood coagulation.

 Thank you very much for your kind comments and indeed you are quite right. We thought we were clear, but indeed it is important to indicate the gross ash content of the feed in the box report. We have it in line 160.  You are absolutely right also about the effect of butyric acid on the other factors.

Minor correction: line 186, Vitamin B12 is technically not a trace element.

Thank you, the change has been made on line 191. Best regards.

Reviewer 2 Report

General points about the case report: This case report brings interesting information regarding the Aluminium intoxication in ruminants. A total of 50 cows presenting abnormal symptoms were investigated and 10 of them were compared to other 10 healthy cows. The case report is very well written, with sufficient information in the different sections. However, I kindly pointed in this letter some issues that must be reviewed and fixed by the authors, or at least they should explain it better.

Specific considerations:

This case report has a mixture of spelling “aluminum” (without i; American spelling) and “aluminium” (with i; British spelling). Please just consider one and fix throughout. I would kindly suggest the British spelling.

Keywords: For indexing reasons, do not use as keywords those words already mentioned in the title. Choose as keywords, different words that those already cited in the title. In this case, I kindly suggest replacing the word “aluminium”, which is already in the title, by something else relevant to this case report.

In this care report, the section “2. Case history”, there is a mixture of description of methodology and results going back and forth throughout this section. Also, there is plenty of discussion in this section and also in the “3. Follow up section”. I would kindly suggest the authors to use a logical order in a way that it would improve readability. For example: Please clearly describe the methodology for data obtention in this case report, then describe the results without discussion; last but not least, make the discussion only in the appropriate section “4. Discussion”.

L97: As well as in L101, add the city and country.

L101: Delete the space before comma.

L103: Here the abbreviation “USA” was used, however in the previous (L101), it was spelled out. Standardize and mention always in the same way.

L104: Delete the space before comma.

L119: In some places it’s written “ULiege”, in some other places it’s “Uliege”. Lower case or capital letter L. Please check throughout and fix it accordingly.

Page 3, paragraph under Figure 1: This is a very long paragraph. I would kindly suggest the authors to divide it at least in 2 paragraphs. Also, this paragraph has a mixture of methodology (performed analyses) and the findings (results) of this case report. Please add them (methodology & results) in different paragraphs, it can potentially improve the readability.

Include a brief description of the statistical analyses in the methodology part. Statistical evaluation from the title of the Figure can then be deleted.

Delete line 123.

L124: Add space after semi colon.

L125: Delete space before comma.

L126: Delete space before semi colon in 2 places.

Please check and fix the space (or lack of space) in this footnote.

Zinc is repeated twice in this footnote.

L139-L140: Please describe UFL and PDI. Check throughout the case report, if abbreviations are missing the description.

L152: Delete space.

L153: Should there be space or not between the number and the %? Please check throughout.

Delete line 156.

L158: “The majority of cows (6/10) of the Sick Pool is under the blue line” this sentence belongs to the results, but not necessarily to the title of Figure. It can be deleted.

L176: Number 3 should be subscript.

First mention Table 2 in the text, and then include Table 2.

Check spaces (or lack of them) in footnotes of Table 2.

L197: Should bold be removed?

L230: Should bold be removed?

L241: The reference between parenthesis has been already mentioned. No need to mention again.

L255: Is “geographically” better and more appropriate than “geologically”?

Should bold be removed from Conclusions section?

L304: Is “silage” a better term than “forage” in this context?

L319: Please make sure that all involved parties, for example farmers, have provided a consent statement.

L321: This sentence is missing an end dot.

L323: Should you properly mention the names and correctly address the acknowledgement? Make sure to have a consent statement from all of them.

Best regards.

Author Response

Please find the answers in the attached document.

Best regards,
